# Global Dynamics of Gastrointestinal Colonisations and Antimicrobial Resistance: Insights from International Travellers to Low- and Middle-Income Countries

**DOI:** 10.3390/tropicalmed9080182

**Published:** 2024-08-17

**Authors:** Laura Seijas-Pereda, David Carmena, Carlos Rescalvo-Casas, Marcos Hernando-Gozalo, Laura Prieto-Pérez, Juan Cuadros-González, Ramón Pérez-Tanoira

**Affiliations:** 1Department of Microbiology, Príncipe de Asturias Universitary Hospital, Carretera Alcalá-Meco s/n, 28805 Alcalá de Henares, Spain; carlos.rescalvo@uah.es (C.R.-C.); m.hernando@uah.es (M.H.-G.); juan.cuadros@uah.es (J.C.-G.); ramon.perezt@uah.es (R.P.-T.); 2Department of Biomedicine and Biotechnology, Faculty of Medicine, University of Alcalá, C/19 Av. De Madrid, 28871 Alcalá de Henares, Spain; 3Parasitology Reference and Research Laboratory, Spanish National Centre for Microbiology, Health Institute Carlos III. Ctra. De Pozuelo, 28, 28222 Majadahonda, Spain; dacarmena@isciii.es; 4Center for Biomedical Research Network (CIBER) in Infectious Diseases, Health Institute Carlos III. C/Merlchor Fernández Almagro, 3, Fuencarral-El Pardo, 28029 Madrid, Spain; 5Department of Organic and Inorganic Chemistry, University of Alcalá, Ctra. Madrid-Barcelona, Km 33, 100, 28805 Alcalá de Henares, Spain; 6Department of Infectious Diseases, IIS-Fundación Jiménez Díaz, Av. De los Reyes Católicos, 2, Moncloa-Aravaca, 28040 Madrid, Spain; lprietope@fjd.es

**Keywords:** gastrointestinal microorganisms, diarrhoeagenic strains, colonisation patterns, infection control, *Escherichia coli* (*E. coli*), beta-lactamase genes (ESBL)

## Abstract

Gastrointestinal microorganism resistance and dissemination are increasing, partly due to international travel. This study investigated gastrointestinal colonisations and the acquisition of antimicrobial resistance (AMR) genes among international travellers moving between Spain and low- and middle-income countries (Peru and Ethiopia). We analysed 102 stool samples from 51 volunteers collected before and after travel, revealing significantly higher rates of colonisation by both bacteria and protists upon return. Diarrhoeagenic strains of *E. coli* were the most notable microorganism detected using RT-PCR with the Seegene Allplex™ Gastrointestinal Panel Assays. A striking prevalence of β-lactamase resistance genes, particularly the TEM gene, was observed both before and after travel. No significant differences in AMR genes were found between the different locations. These findings highlight the need for rigorous surveillance and preventive strategies, as travel does not significantly impact AMR gene acquisition but does affect microbial colonisations. This study provides valuable insights into the intersection of gastrointestinal microorganism acquisition and AMR in international travellers, underscoring the need for targeted interventions and increased awareness.

## 1. Introduction

Gastrointestinal infections and antimicrobial resistance (AMR) pose a significant global health challenge due to their ubiquity and complexity of treatment. The prevalence and types of gastrointestinal infections and resistance can vary greatly across regions, with low- and middle-income countries (LMICs) experiencing a greater impact. Notably, Asia bears the largest burden of AMR, closely followed by North Africa and other countries within the Global South [1,2]. This greater AMR burden can be attributed to the unique living conditions and challenges associated with antimicrobial usage in these regions. Extensive antibiotic use in humans, animals, and agriculture, coupled with inadequate healthcare infrastructure and sanitation facilities, as well as the forces of globalisation, drive the spread and the increase of microbial colonisation. This can lead to the development of AMR and complicated infections that transcend geographical boundaries [3,4,5].

Given the enormous growth of international travel, there is a rising concern about the potential repercussions on the acquisition and dissemination of certain microorganisms and their microbial resistances worldwide. Resistant bacteria and the elements carrying these resistances (e.g., plasmids) can be rapidly transported across regions [6]. The main AMR mechanisms identified are extended-spectrum β-lactamase (ESBL) enzymes and carbapenemases, which confer resistance to a broad spectrum of β-lactam and carbapenem antibiotics, respectively [1,7]. Furthermore, these challenging bacteria often exhibit resistance to multiple classes of antibiotics, limiting the availability of effective antimicrobial agents for infection prevention and treatment.

The specific types of gastrointestinal microorganisms responsible for infections and the prevalence of AMR genes exhibit regional disparities, indicating that the destination of travel is a strong predictor of the acquired microorganisms and the extent of their resistance [8,9]. Several studies have investigated the prevalence of different ESBL-producing bacterial strains. For instance, in Enterobacteriaceae such as *Escherichia coli*, the highest consistent resistance rates of 20–70% have been reported in various Asian countries, while rates are described as 10–15% in South America and 10–40% in Africa (except for *E. coli* in Egypt, where ESBL-producing strains have been reported to exceed 60%). In contrast, significantly lower prevalence rates have been seen in most European countries (2–12%) and North America (<3%), primarily due to more restrictive antibiotic use in humans and animals [3,9,10]. Therefore, the movement of travellers from areas with higher prevalence to those with lower prevalence can contribute to the transmission of these resistant microorganisms, potentially affecting the management of future gastrointestinal infections.

Given the escalating threat of AMR in both healthcare facilities and community settings, with consequential negative impacts on health and economics, the importance of maintaining vigilant supervision and containing its spread cannot be overstated [11,12]. There is an increasing demand for research in this field to support targeted interventions and screenings for international travellers, not only when symptoms of disease are evident but also among asymptomatic individuals who may have been colonised. This study aims to investigate gastrointestinal colonisations and the acquisition of AMR among international travellers moving between regions with varying AMR prevalence. Through this research, we intend to establish a foundation for direct interventions and surveillance measures to mitigate the spread of AMR.

## 2. Materials and Methods

### 2.1. Study Design and Sample Collection

This retrospective observational cohort study was conducted between 2021 and 2023. The study population comprised Spanish master’s students living in Madrid, who travelled to urban and peri-urban areas of Iquitos (Peruvian Amazon) or rural areas of Gambo (West Arsi, Ethiopia) as part of their educational programme. These students actively participated in various healthcare centres and hospitals during short stays of approximately 23 days. Participants voluntarily provided individual stool samples both before departure (collected in Spain) and upon their return to Madrid, Spain (collected in Peru or Ethiopia). Stool samples were preserved in sterile containers with 70% ethanol and kept refrigerated before undergoing processing and analysis in the laboratory.

### 2.2. Sample Processing

Stool samples were processed at the microbiology laboratory of the *Príncipe de Asturias University Hospital* in Alcalá de Henares, Spain. The analysis began with DNA extraction using the STARMag universal cartridge kit (Seegene Inc., Seoul, Republic of Korea) on the Microlab STARTlet Hamilton platform (Werfen, Barcelona, Spain). Subsequently, gastrointestinal microorganisms were investigated using polymerase chain reaction (PCR) with three multiplex molecular panels Allplex™ GI Bacteria (I-II) and GI Parasite Assay (Seegene Inc., Seoul, Republic of Korea) for the detection of the following potential pathogens:GI-Bacteria (I) Assay: *Aeromonas* spp., *Campylobacter* spp., *Clostridioides difficile* toxin B, *Salmonella* spp., Enteroinvasive *Escherichia coli*/*Shigella* spp., *Vibrio* spp., and *Yersinia enterocolitica*.GI-Bacteria (II) Assay: Enteroaggregative *E. coli* (EAEC), Enteropathogenic *E. coli* (EPEC), *E. coli* O157, Enterotoxigenic *E. coli* (ETEC), Hypervirulent *Clostridioides difficile*, and Enterohemorrhagic *E. coli* (EHEC).GI-Parasite: *Blastocystis hominis*, *Giardia lamblia*, *Dientamoeba fragilis*, *Entamoeba histolytica*, *Cyclospora cayetanensis*, and *Cryptosporidium* spp.

Additionally, the presence of key AMR genes was investigated using three different VIASURE Real-Time PCR Detection Kits (Certest, Zaragoza, Spain) for the following:ESBL genes: TEM, SHV and CTX-M, and Colistin resistance gene: mcr-1.Vancomycin resistance genes: VanA and VanB.Carbapenemase-encoding genes: NDM, VIM, OXA, KPC, and IMP.

In every PCR analysis, samples with cycle threshold (CT) values lower than 38 were considered positive.

### 2.3. Statistical Analysis

Statistical analyses were performed using STATA/MP 17.0 (StataCorp, College Station, TX, USA). Continuous variables were expressed as medians with interquartile ranges (IQR), while categorical variables were presented as proportions unless otherwise specified. Group differences were assessed using appropriate statistical tests, including the Mann–Whitney U test for continuous variables and the χ^2^ test or Fisher’s exact test for categorical variables. All *p*-values were calculated two-tailed, with significance defined as a *p*-value ≤ 0.05.

## 3. Results

A total of 102 individual stool samples were collected, comprising one before departure and one upon arrival from each of the 51 volunteers. Inbound and outbound stool samples were collected within two days of departure and arrival, respectively. Participants’ ages ranged from 22 to 65 years, with a female-to-male ratio of 2:1. Sixty-six (64.7%) samples (from 33 individuals) were collected from the Ethiopia trip, and thirty-six (35.3%) samples (from 18 individuals) were obtained from the Peru trip. Participants did not report any specific symptoms or antibiotic intake during their stay abroad.

Overall, 59.8% (61/102) of the samples analysed by PCR tested positive for one or more of the investigated gastrointestinal microorganisms. Of these, 65.6% (40/61) were of bacterial nature, and 54.1% (33/61) were of parasitic nature. Among the samples collected prior to departure, 43.1% (22/51) of the participants tested positive for either bacterial or protist microorganisms. After arrival, this frequency increased to 76.5% (39/51), a statistically significant difference (*p*-value = 0.001). Interestingly, volunteers returning from their trip to Peru were significantly more likely to acquire infections or colonisations (94.4%, 17/18) by the surveyed microorganisms than those travelling to Ethiopia (66.7%, 22/33) (*p*-value = 0.007). More bacteria were detected (52.9%, 27/51) than parasites (39.2%, 20/51) among all returning travellers, a trend also significant in both countries: Ethiopia (42.4%, 14/33) with a *p*-value of 0.008 and Peru (72.2%, 13/18) with a *p*-value of 0.019. Gender was not identified as a factor associated with a higher likelihood of gastrointestinal colonisations (*p*-value = 1.000).

Focusing on the investigated bacteria, EPEC was the most frequently identified potential pathogen in the stool sample panel [27.5%, 28/102; 95% confidence interval (CI): 19.1–37.2], followed by EAEC (12.8%, 13/102; 95% CI: 7.0–20.8), ETEC (11.8%, 12/102; 95% CI: 6.2–19.7), *Aeromonas* spp. (4.9%, 5/102; 95% CI: 1.6–11.1), *Campylobacter* spp. (2.0%, 2/102; 95% CI: 0.2–6.9), and *Clostridioides difficile* toxin B and EHEC (1.0%, 1/102; 95% CI: 0.02–5.3 each). Among protists, the most frequent species found was *B. hominis* (28.4%, 29/102; 95% CI: 19.9–38.2) followed by *D. fragilis* (7.8%, 8/102; 95% CI: 3.5–14.9) and *G. lamblia* (2.9%, 3/102; 0.6–8.4). Coinfections of bacteria and protists were identified in 11.8% (12/102) of samples, and 22.5% (23/102) of samples harboured two or more type of bacteria. Importantly, none of the 102 stool samples examined tested positive for *Salmonella* spp., enteroinvasive *Escherichia coli*/*Shigella* spp., *Vibrio* spp., *Yersinia enterocolitica,* hypervirulent *Clostridioides difficile*, *Entamoeba histolytica, Cyclospora cayetanensis*, or *Cryptosporidium* spp.

Table 1 shows the rates of positive microorganisms isolated from the international travellers before departure and upon arrival. Colonisation rates before travel were relatively low for both bacterial (range: 0–15.7%) and protist (3.9–21.6%) microorganisms, with a general increase observed after arrival (ranges: 0–39.2% for bacteria and 2.0–35.3% for protists). Notably, colonisation rates for EAEC and EPEC were significantly higher in returning travellers from LMICs (*p*-values = 0.002 and 0.014, respectively).

We also investigated the presence of AMR genes in a subset of 76 samples, comprising 38 collected before departure and 38 upon return (Table 2). Among them, 42 (55.3%) were from travellers returning from Ethiopia, and 34 (44.7%) were from travellers returning from Peru. Remarkably, all samples except two (97.4%, 74/76) tested positive for at least one resistance gene. Interestingly, none of the samples investigated tested positive for VanA or NDM resistance genes. Broadly speaking, β-lactamase resistance genes (TEM, CTX, and SHV-1) were highly prevalent both before and after travel, with only three samples testing negative for them. In contrast, carbapenemase resistance genes (VIM, OXA, IMP, and KPC) were far less common, with only five positive samples. Our analyses revealed no statistically significant differences in the presence of any resistance gene before and after travel (*p*-value = 0.493). Similarly, the rates of resistance-gene presence were not associated with gender (*p*-value = 0.535) or with the returning country (Ethiopia or Peru, *p*-value = 0.499).

In light of the preceding observations, we analysed the relationship between a positive PCR result for any of the studied microorganisms (bacteria and protists) and the carriage of AMR genes, discerning no significant association (*p*-value = 1.000). When examining this relationship considering only the presence of the bacteria analysed, our investigation also revealed no statistically significant correlation (*p*-value = 1.000). 

## 4. Discussion

In this study, we investigated the variable presence of gastrointestinal microorganisms and AMR genes among international travellers moving between regions with different prevalence levels (Spain, Ethiopia, and Peru). The most notable contributions of the study to the field are the demonstration that returning travellers from LMICs had a significantly higher rate of colonisation by both bacterial and protist microorganisms, and the high carriage rate of AMR genes in the study population, despite their travels to LMICs.

Among the 51 stool samples analysed before departure, the prevalence of different intestinal microorganisms ranged from 2% for EAEC to 21% for *B. hominis*. EPEC was the most frequently detected bacterium, with a prevalence comparable to that reported in other studies conducted in the Madrid population [13,14]. Some common pathogens, such as *Salmonella* spp., were not identified in this study. This may be attributable to the relatively small number of stool samples available but also to their epidemiology and relation with symptomatic infection, which was not always the case in our cohort [15,16]. The prevalence of gastrointestinal microorganisms clinically relevant in LMICs has been described in many global studies, where reported rates over 20% are common. This is a direct consequence of limited access to safe drinking water and food, poor hygiene conditions, insufficient sanitary facilities, and weakly implemented antimicrobial policies in these regions [17,18]. When people travel to and live in these settings, the acquisition of some of these microorganisms is highly likely, with implications not only for the health of the affected individuals but also for the potential transmission and impact upon the travellers’ return [1,19,20].

When comparing inbound and outbound samples, we found significantly higher positivity rates for both bacterial and protist microorganisms (up to 76.5%) among returning travellers. This is consistent with findings from previous studies [5,11], where the acquisition of gastrointestinal microorganisms during travel and residence in LMICs appears to be commonplace. Therefore, there is a pressing need for larger-scale investigations involving both healthy travellers and those exhibiting symptoms, to comprehensively evaluate the genuine risks and implications associated with these findings and to determine accurately whether these instances constitute a risk for home country populations.

Although travellers to both Ethiopia and Peru exhibited high rates of positivity, it is worth noting that we observed differences between the two destinations, with returning samples from Peru showing a statistically higher rate of positive isolates. Most studies confirming higher colonisations and infection rates in LMICs typically report that Asia and Africa have the highest colonisation rates upon travellers’ return [6,13], likely due to more frequent investigation. In the present study, it appears that travellers staying in rural South America (Iquitos, Peru) are more affected than those in the countryside of Africa (Gambo, Ethiopia), a trend supported by some existing studies [21]. This may be influenced by several factors such as the state of water and food consumed, the type of travel, the specific place where the traveller stayed, and other environmental conditions [10,11]. Furthermore, we identified a notable number of isolates from diarrhoeagenic *E. coli*, specifically enteroaggregative and enteropathogenic strains. The presence of these diarrhoeagenic *E. coli* strains was somewhat expected, as they are widely reported as the primary cause of diarrhoea in both children and travellers from LMICs, where the presence of these microorganisms and their significant impact on health have been well-documented in various studies [17,22,23].

On the other hand, our study reveals alarming carriage rates of resistance genes across all the samples studied, particularly concerning β-lactamases. The prevalence in European and North American community settings is reported to be quite low (<10%) despite recent reports raising concerns about the steadily increasing global prevalence of β-lactamase genes in the community [24,25]. With an annual increase of up to 5%, international travellers are identified as a particularly high-risk population in this regard [1,3]. Notably, our study demonstrates an exceptionally high prevalence of the TEM gene, with rates as high as 96%, exceeding what is typically reported in the literature [4,24]. Moreover, we observed similar prevalence rates in Spain and among travellers returning from LMICs (Peru and Ethiopia) (*p*-value = 0.493). This suggests that travelling to these countries does not have a discernible impact on β-lactamase gene colonisation and highlights the substantial presence of these resistances in the community within more developed countries, such as Spain. In this context, human-to-human transmission gains importance [26]. Although antibiotic use and travellers’ diarrhoea have been shown to influence the risk of ESBL acquisition, the specific mechanisms underlying successful colonisation during travel and its long-term implications remain largely unknown [12,20,27]. In contrast, we observed the anticipated rates of resistance genes for colistin (mcr-1) and vancomycin (VanA, VanB) [28]. Our study documented a very low prevalence of carbapenemase-resistance genes (NMD, VIM, OXA, KPC, IMP), with only five positive cases, which is consistent with findings reported in many other studies [19,29,30]. Nevertheless, it is important to acknowledge that carbapenemases are continually evolving, and the global spread of mobile genes encoding carbapenem-resistant traits has led to infections associated with increased morbidity and mortality [31,32,33].

The limitations of our study include its single-centre nature and retrospective design. Sample collection was dependent on the voluntary participation of the recruited students, which may introduce some bias into the results. Additionally, the study involved a relatively small cohort; a larger sample size could have provided more precise data on the occurrence and frequency of gastrointestinal bacterial and protist microorganisms, as well as the carriage of antimicrobial resistance (AMR) genes. It is important to recognise that our study population may not be fully representative of all travellers, given their specific roles in providing assistance at hospitals and healthcare centres. Therefore, the data presented should not be extrapolated to other geographical regions with differing epidemiological and socioeconomic contexts. Lastly, our study employed only molecular methods for detecting specific nucleic acid sequences and did not include an analysis of gastrointestinal viruses. Future research should address this gap.

## 5. Conclusions

This study illuminates the complex dynamics of gastrointestinal colonisations and the acquisition of antimicrobial resistance among international travellers. It may be summarised as follows:-Elevated rates of positive isolations: Significantly higher rates of positive isolations for both protozoal and bacterial microorganisms were observed in travellers returning from LMICs, specifically Peru and Ethiopia. This highlights an urgent need for further research, as the full implications of these findings remain unclear.-Prevalent diarrhoeagenic *E. coli* strains: Enteroaggregative and enteropathogenic strains of *E. coli* emerged as the most common microorganisms among returning travellers.-Comparative risk by destination: Travel to Peru was associated with a higher likelihood of colonisations by gastrointestinal bacteria and protists compared to travel to Ethiopia. However, larger-scale studies are necessary to confirm this trend.-AMR gene prevalence: No significant differences were found in the prevalence of antimicrobial resistance (AMR) genes between pre- and post-travel samples, underscoring the extensive global prevalence of these genes within communities.-Concerning β-lactamase trends: The high prevalence of β-lactamase genes, particularly the TEM gene, reflects a troubling global trend in antimicrobial resistance. Our findings suggest that β-lactamases are widespread in the community, with transmission potentially occurring through both travel and human-to-human contact.

This study underscores the critical need for targeted interventions and heightened awareness among travellers. It highlights the importance of robust surveillance, continued research, and strategic actions to mitigate the spread of AMR and gastrointestinal colonisations among international travellers and within affected communities. Future research involving healthy travellers is crucial to fully understand the risks and implications of these findings on a global scale.

## Figures and Tables

**Table 1 tropicalmed-09-00182-t001:** Rates of colonisation by gastrointestinal microorganisms in international travellers (*n* = 102) before departure to and upon arrival from low- and middle-income countries. Statistically significant values (*p*-value ≤ 0.05) are bolded.

Gastrointestinal Microorganisms	Before (*n* = 51)	After (*n* = 51)	*p*-Value
**Bacteria**			
*Aeromonas* spp.	3 (5.9%)	2 (3.9%)	1.000
*Campylobacter* spp.	0 (0%)	2 (3.9%)	0.495
*Clostridioides difficile* toxin B	0 (0%)	1 (2.0%)	1.000
Enteroaggregative *E. coli* (EAEC)	1 (2.0%)	12 (23.5%)	**0.002**
Enteropathogenic *E. coli* (EPEC)	8 (15.7%)	20 (39.2%)	**0.014**
Enterotoxigenic *E. coli* (ETEC)	3 (5.9%)	9 (17.6%)	0.122
*E. coli* O157	0 (0%)	1 (2.0%)	1.000
Enterohemorrhagic *E. coli* (EHEC)	1 (2.0%)	0 (0%)	1.000
**Parasites**			
*Blastocystis hominis*	11 (21.6%)	18 (35.3%)	0.187
*Dientamoeba fragilis*	3 (5.9%)	5 (9.8%)	0.715
*Giardia lamblia*	2 (3.9%)	1 (2.0%)	1.000

**Table 2 tropicalmed-09-00182-t002:** Carriage rates of antimicrobial resistance genes in a subset of international travellers (*n* = 76) before departure to and upon arrival from low- and middle-income countries. PCR-positive results are indicated.

Detected Gene	Before (*n* = 38)	After (*n* = 38)	Total Positive PCR (*n* = 76)	*p*-Value
CTX	16 (42.1%)	23 (60.5%)	39 (51.3%)	0.168
TEM	37 (97.4%)	36 (94.7%)	73 (96.1%)	1.000
SHV-1	16 (42.1%)	24 (63.2%)	40 (52.6%)	0.107
mcr-1	3 (7.9%)	3 (7.9%)	6 (7.9%)	1.000
VanB	12 (31.6%)	7 (18.4%)	19 (25.0%)	0.289
VIM	0 (0.0%)	1 (2.6%)	1 (1.3%)	1.000
OXA	1 (2.6%)	0 (0%)	1 (1.3%)	1.000
KPC	1 (2.6%)	0 (0%)	1 (1.3%)	1.000
IMP	1 (2.6%)	1 (2.6%)	2 (2.6%)	1.000

## Data Availability

The data presented in this study are available on request from the corresponding author due to privacy reasons and ethical restrictions.

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
