# Peer review of "Global Dynamics of Gastrointestinal Colonisations and Antimicrobial Resistance: Insights from International Travellers to Low- and Middle-Income Countries"

_tropicalmed, 2024, doi:10.3390/tropicalmed9080182_

Round 1

Reviewer 1 Report

Comments and Suggestions for Authors

1. The authors describe the transmission of gastrointestinal infections and AMR genes from Low income countries to high income countries through travelers. English language editing is required. Some sentences are difficult to comprehend. lines 79-81, meaning is not clear.

2. The method of detection used here should be described. 

3. References are not up-to-date.

4. Authors have not mentioned about ethical considerations and informed consent for the study? Was it obtained? Authors should include that in the manuscript.

5. Authors have not mentioned in the aticle the sample collection procedure, mode of samples transport and adequatelte storage facilities, which should be included in the methods.  More statistical analysis is necessary to support their findings. The method used for detection has not been described adequately. Controls are not included. Materials and Methods are very poorly presented.

6. Results before and after travel must be presented in tables to include demographic details like age, sex, gender etc.  Authors has made only a country based analysis. More granular data at the field level would have strengthen the analysis of the article. They should mention which part of the countries the traveler had visited and the part of Spain where did the person belong.

7. In Discussion  (lines 235-236) authors showed that travelers to S. America were more affected than to Africa. Did it indicate any public health trend? Authors need to elaborate this in the discussion with more analysis of current data.

8. Diarrheagenic E. coli was the main pathogen detected. Please mention their prevalence in Spain? Authors should mention this in the Discussion and compare the results with the other two destination countries.

9. Conclusion must be improved.

10. Organism names must be in italics.

Comments on the Quality of English Language

Grammatically okay , however, in some places the sentences are difficult to comprehend due to complicated presentation.

Author Response

All comments have been addressed in the attached PDF.

Thank you and best regards,

Seijas et al.

Reviewer 2 Report

Comments and Suggestions for Authors

The subject of gastrointestinal infections related to travel is extensively studied. The authors performed a study including 102 individuals who traveled to Peru and Ethiopia. Stool samples were collected before and after travel and analyzed for intestinal pathogens and AMR genes.

Almost 60% of patients tested positive by the PCR tests. However, there is an important difference between colonization and infection that is not clearly presented in the manuscript. Of the 102 subjects, how many had symptoms after return? If none, then the study is analyzing the colonization, not the infections and the title is inappropriate. Also, the aim is, partly, to “illuminate” (strong word?) “the implications of gastrointestinal infections”, but were there indeed infections?

Line 73: Please classify why was K. pneumoniae mentioned.

2.2. Sample Processing: The authors mention the amplification kits used for PCR. How were the nucleic acids extracted?

Lines 119-194: Please provide more details about how the resistance genes were detected. Are the mentioned targeted genes included in a multiplex panel or were they individually detected? Is/Are the kit(s) approved for stool samples?

Line 155: “EPEC was the most frequent pathogen”. EPEC has clinical significance only on children aged <1 year old. The subjects included in the study are master students, >20 years old. The clinical significance of these findings has to be discussed in the “Discussion” section.

Lines 155-165: are these results from samples collected before or after return?

Lines 204-208 present the most important contribution of the study. The authors state that “returning travellers from LMICs had 205 a significantly higher rate of infection/colonization by both bacteria and protist pathogens”, but in Table 2 the statistically significant results are for EPEC (without importance for students) and EAEC alone. Also, the conclusions (especially lines 288-289) are not fully supported by the results.

Line 275-285: Limitations of the study: I enjoyed the paragraph. Please also add the limitations of the used methods, as the PCR only detects specific sequences of nucleic acids.

Lines 296-297: The magnitude of ESBL genes in the studied population?

The last paragraph of the conclusion mentions again “gastrointestinal infection”. Please clarify in the entire manuscript if these are cases of infection of colonization and use the terms appropriately.

Comments on the Quality of English Language

Minor language revisions are advised. 

Author Response

(The authors gave the same response as above.)

Reviewer 3 Report

Comments and Suggestions for Authors

Title: Global Dynamics of Gastrointestinal Infections and Antimicrobial Resistance: Insights from International Travellers to Low- and Middle-Income Countries

 Line 139 : Authors can provide a Table containing the information (age, gender etc.) the participants of the study.

Did the authors observe any relation between the pathogens detected or the number of  AMR genes detected and the length of stay of the travellers?

In line 97, the authors mentioned that the participants actively participated in various healthcare centres and hospitals during short stays ranging from 15 to 30 days.  As the travellers were engaged in activities in healthcare settings, the authors should revise line 217: ‘Our findings suggest that ß-lactamases are deeply rooted in the community and may not necessarily be impacted by travel, but also of risk, emphasizing the critical role of human-to-human transmission.  ”  to avoid confusion. Authors have recruited participants involved in health care settings and in the discussion they mentioned that their finding suggests that the ‘ß-lactamases are deeply rooted in the community’.

Authors can discuss the prevalence rate of AMR in Peru, Ethiopia and Spain.

Author Response

(The authors gave the same response as above.)

Reviewer 4 Report

Comments and Suggestions for Authors

The work is interesting but I think that it presents some big question:

1: DNA methods underline that also dead microorganisms can be identified, particularly parasites...it is necessary to perform O&P gold standard method to detect the presence of parasites

2: How is it possible to detect before vancomycin resistant bacteria and after the travel the number was decreased??

3: Lack of data about treatment or antibiotic therapy during the staying in the other states...very important to obtain a confirmation of the data in the results!!!

Comments on the Quality of English Language

Minor English revision, some mistakes throughout the manuscript

Author Response

(The authors gave the same response as above.)

Round 2

Reviewer 2 Report

Comments and Suggestions for Authors

Thank you for addressing my concerns. There is a clearer destining between colonization and infection now and the manuscript improved.

Line 72: Does the references only include the “Enterebacteriaceae” family, or would it be better to refer to the Enterobacterales order?

Comments on the Quality of English Language

The English is fine.

Reviewer 4 Report

Comments and Suggestions for Authors

Thank you for your reviosn, I am looking for the next future research concerning O&P method plus Moleculari biology tecnique.